# Deltamethrin’s Effect on Nitrogen-Fixing Nodules in *Medicago truncatula*

**DOI:** 10.3390/toxics12080615

**Published:** 2024-08-21

**Authors:** Rosario De Fazio, Cristian Piras, Domenico Britti

**Affiliations:** 1Department of Health Sciences, “Magna Græcia University” of Catanzaro, Campus Universitario “Salvatore Venuta” Viale Europa, 88100 Catanzaro, Italy; britti@unicz.it; 2CISVetSUA, “Magna Græcia University” of Catanzaro, Campus Universitario “Salvatore Venuta” Viale Europa, 88100 Catanzaro, Italy

**Keywords:** deltamethrin, pyrethroid, nitrogen-fixing nodules, soil fertility, soil health, livestock farming, green veterinary pharmacology

## Abstract

Deltamethrin is used against plant pests (e.g., mites and ants) and, in farm animals, against biting insects because of its acaricidal and repellent effects against ticks, thus protecting the sheep and cattle from the transmission of pathogens. However, its impact on the environment still needs to be fully evaluated. This study evaluates the impact of this pyrethroid on the nitrogen-fixing nodules in *Medicago truncatula*, a model legume. This research compares nodular biomass and root weight between a deltamethrin-treated section and an untreated control section of this legume. Our results indicate a significant reduction in the biomass of nitrogen-fixing nodules in the treated grove, suggesting that deltamethrin negatively affects the symbiotic relationship between *M. truncatula* and nitrogen-fixing bacteria. This reduction in nodule formation can impair soil fertility and plant growth, highlighting an ecological risk associated with pyrethroid’s use in livestock farming. These findings underscore the need for a shift towards Green Veterinary Pharmacology (GVP), which promotes environmentally sustainable practices in managing livestock health. By minimizing our reliance on harmful chemical treatments, GVP offers viable solutions to protect and enhance ecosystem services such as biological nitrogen fixation that are essential for maintaining soil health and agricultural productivity.

## 1. Introduction

The widespread use of chemical treatments in agriculture and livestock farming has raised significant concerns about their environmental impacts [1].

Deltamethrin is applied to crops such as cotton, vegetables, and fruits, both as a foliar spray and in soil treatments. It is effective against insects like aphids, caterpillars, and beetles as it disrupts their nervous system, leading to paralysis and death. Moreover, it is extensively employed to protect farm animals, particularly sheep and cattle, from biting insects and ticks [2]. These ectoparasites not only cause distress and economic losses by reducing productivity but also act as vectors for various diseases [3]. Deltamethrin’s efficacy in controlling these pests is attributed to its strong acaricidal and repellent properties, making it a popular choice among farmers and breeders [2]. However, the extensive application of deltamethrin for any of these described purposes poses potential ecological risks, particularly concerning its impact on non-target organisms and crucial soil processes [1,2,3,4].

Fly infestations are a major issue in dairy cattle herds due to their rapid reproduction, especially in warm, humid conditions. Flies cause severe welfare and productivity problems by biting, causing skin lesions, and disrupting feeding and relaxation. They also spread diseases like keratoconjunctivitis and mastitis.

Their effective control involves reducing breeding areas through hygiene practices and using insecticides and repellents. Deltamethrin is highly effective in cattle due to its “hot foot effect”, killing flies on contact [5]. On the other hand, the primary route for excreting the absorbed amount of deltamethrin in the target animal is through the feces. Through this, deltamethrin can potentially harm non-target organisms; after treatment, deltamethrin is expelled in the feces for the next 2 to 4 weeks [6].

Among the non-target organisms threatened there are nitrogen-fixing bacteria that are crucial due their symbiotic action of fixing nitrogen for plants.

In vivo studies have revealed that certain organochlorine pesticides, agrichemicals, and environmental contaminants disrupt the symbiotic relationship between rhizobia bacteria and host plant roots. This interference results in inhibited or delayed rhizobia recruitment, fewer root nodules, diminished nitrogenase activity, and, ultimately, a reduction in overall plant yield at harvest. The environmental repercussions of such disruptions include increased reliance on synthetic nitrogenous fertilizers, diminished soil fertility, and unsustainable long-term crop yields [7].

Nitrogenase, the enzyme responsible for fixing atmospheric nitrogen gas (N_2_), is particularly affected by pesticide applications, which reduce its efficiency and activity. Previous studies have reported a decrease in total nitrogenase activity in pots sown with Pisum sativum plants following herbicide applications [8].

Moreover, it was demonstrated that some pesticides interfere with the NodD receptors in *Sinorhizobium meliloti* bacteria, which are similar to estrogen receptors. These receptors respond to the flavonoid molecules produced by plants to form nodules and fix nitrogen. Pesticides reduce the NodD signal, compromising nitrogen fixation [9].

*Medicago truncatula*, a model legume [10], plays a pivotal role in agricultural systems due to its ability to form symbiotic relationships with nitrogen-fixing bacteria (*Rhizobia*). This symbiosis results in the formation of nodules on the plant roots, where atmospheric nitrogen is converted into ammonia, a form usable by plants [11]. This biological nitrogen fixation is crucial for soil fertility and plant growth, reducing the need for synthetic fertilizers and enhancing sustainable agricultural practices [12].

From this perspective, deltamethrin’s effect on the nitrogen-fixing ecosystem has still not been clearly evaluated. Its effect was demonstrated on cyanobacteria on rice [13], but more evidence is needed to better evaluate its effect on the environment.

The aim of this study was to assess whether soil exposure to residues of this synthetic pyrethroid could affect the overall growth of the nitrogen-fixing nodules of *Medicago truncatula*. Understanding how deltamethrin influences this critical ecological interaction is essential for assessing its overall environmental impact.

This study was performed by comparing the normalized nodule biomass between two different sites in the same field (treated vs. non-treated), allowing for the quantification deltamethrin’s impact on the symbiotic relationship between *M. truncatula* and nitrogen-fixing bacteria.

## 2. Materials and Methods

### 2.1. Field Treatment and Sample Collection

The sampling of *M. trunculata* root samples took place in a single olive grove located in Lamezia Terme in Calabria, southern Italy. The olive grove was divided into two parts: one treated and one untreated. The only introduced variable was the periodic application of an insecticide based on deltamethrin (0.0015 g/m^2^) in the treated part of the olive grove. This insecticide was applied three times, at intervals of approximately 21 days, during the period from July to September (Figure 1). Samples were collected by extracting whole soil turfs in the month of May. To prevent airborne soil contamination, control samples were taken at a distance of 50 m from the treated area. In total, 20 samples were collected from each area. The selection of these samples was based on a specific criterion: only plants with a length of approximately 60 cm were considered eligible for inclusion in the sample. This approach ensured that all selected samples had a uniform and representative size, thus facilitating a more accurate comparison of results between the different areas. Additionally, the samples were weighed blindly to ensure the absence of bias or influence in the collected data.

### 2.2. Sample Preparation

Root samples were thoroughly rinsed under running water to remove any soil residues. After this cleaning phase, the roots were spread on a dry surface for one hour for partial desiccation. Once dried, the root nodules were gently separated from the roots using tweezers, ensuring their structural integrity was maintained. Both the pink root nodules and the remaining part of the root were examined [14]. Samples were temporarily stored at −20° and subsequently transported (using dry ice) to the laboratories of Centro Interdipartimentale di Servizi Veterinari, Magna Graecia University of Catanzaro, Italy, for long-term storage (−80°). Subsequently, both the root and root nodule samples were weighed.

### 2.3. Soil Analysis

For the pH analysis, 20 g of air-dried soil was transferred into a water bottle. Forty milliliters of deionized water was added, and the bottle was shaken vigorously for one minute. The solution was allowed to settle for 20–30 min. The pH meter used was then calibrated according to the manufacturer’s instructions. The pH was measured by immersing both electrodes into the supernatant solution, ensuring they did not come into direct contact with soil residues [15].

To determine soil texture using the silt–clay separation method, three soil samples, each passing through a 63 µm sieve, were prepared. Each 50 g sample was soaked in a sodium hexametaphosphate solution for 24 h to disperse clay particles. The mixture was then transferred to modified sedimentation cylinders and filled with distilled water to the 1000 mL mark. The cylinders were shaken for 2 min in a shaker and allowed to settle for 2 h. After sedimentation, a tube connected to the cylinder collected a sample of the liquid 30 mm below the water’s surface, which corresponds to the sedimentation of 2 µm particles according to Stokes’ law. The sample was then dried and weighed to determine the amount of silt and clay in the soil [16].

### 2.4. Statistical Analisys

Once the weights of the roots and radical nodules were recorded, we conducted a statistical analysis. We normalized the weight of the radical nodules to the weight of the roots by calculating the ratio of the weight of the nodules to that of each root. Additionally, a statistical analysis was performed using the total weight of the roots and nodules, the weight of the roots alone, and the weight of the nodules alone.

Subsequently, we performed a statistical analysis using JMP Statistical Discovery (version 14; SAS Institute Inc., Marlow, UK) and using the Kruskal–Wallis test (rank sums), to evaluate the differences between the groups.

## 3. Results

*Medicago truncatula*, known as barrel clover or barrel medick, is a small annual legume native to the Mediterranean that is used in genomic research. This clover-like plant grows 10–60 cm tall with trifoliate leaves and yellow flowers, producing small, spiny pods. It is a model organism for legume biology due to its small diploid genome, rapid generation time, prolific seed production, and amenability to genetic transformation. Unlike *Arabidopsis thaliana*, *Medicago truncatula* forms symbioses with nitrogen-fixing rhizobia and arbuscular mycorrhizal fungi, making it crucial for studying these processes.

As numerous xenobiotics negatively influence nitrogen-fixing rhizobia, we evaluated their presence and weight after deltamethrin exposure.

In Lamezia Terme, the precipitation levels rose significantly from 24 mm in July to a peak of 110 mm in November, indicating increased rainfall towards the end of the year (Table 1) [17]. The maximum temperatures showed a gradual decline from 27 °C in July to 13 °C in December, while the minimum temperatures similarly dropped from 21 °C to 9 °C, reflecting a typical seasonal cooling pattern (Table 2 and Table 3) [17]. The soil in the study area has a pH of 6.8, indicating slightly acidic conditions. It is classified as clayey, containing 35.55% silt, 17.37% sand, and 47.08% clay, suggesting high water retention and potential implications for soil management (Table 4).

In Figure 2, it is possible to see a representation of the difference between the weight of the nitrogen-fixing nodules of the control group and the treated group. The nitrogen-fixing nodules’ weight was higher in the control group, with the difference between the weights being ≥30% (fold change (FC) 0.416) and with a *p* value of 0.0103 (Table 5).

On the other hand, the weight of the root system without nodules was higher in the group of plants exposed to deltamethrin, with a 1.33 fold change and a *p*-value of 0.0104 and, as in Figure 3 and Table 6; the mass of the entire root system (including nitrogen-fixing nodules) was similarly higher (Table 7, Figure 4) in the plants exposed to deltamethrin (fold change 1.216; *p*-value 0.0096).

The normalized weight of the nodules (their ratio with the roots’ weight without nodules) is shown in Figure 5 and clearly demonstrates that the nodules are much smaller (fold change 0.36 and *p*-value equal to 0.0001) in the root system of the plants treated with deltamethrin (Table 8).

The results are derived from the data presented in Appendix A.

## 4. Discussion

The focus of this research study was to define the role of a pyretrhoid commonly used against flies and ticks in animals and against parasites in crops on the nitrogen-fixing nodules of *Medicago truncatula*.

Deltamethrin is widely used in the animal field for controlling various parasites that infest livestock. In the field of plants, is approved for application on a wide range of crops such as cotton, corn, cereals, soybeans, and various vegetables. It is utilized to combat a diverse array of agricultural pests, including, but not limited to, mites, ants, weevils, and beetles.

This compound is effective in repelling and killing arthropods such as flies, ticks, lice, and mites. Scientific studies have shown that deltamethrin protects animals for over 4–5 weeks, even under wet conditions. It is used to protect ruminants from midges and other insects, reducing the risk of diseases transmitted by these parasites. It is effective against ticks in all their developmental stages and against mosquitoes. Its available formulations include sprays, medicated baths, and spot-on treatments. Therefore, deltamethrin is an important tool in managing animal health, contributing to the welfare of livestock and the productivity of farms [18].

After livestock’s treatment with pyrethroids, the main route of deltamethrin’s excretion is through feces, with detectable residues in dung for up to two weeks post-application. Topical formulations are the primary source of dung contamination, with a significant portion of the administered dose found in feces following deltamethrin use. Pyrethroids degrade slowly in dung, with studies indicating that concentrations of deltamethrin remain constant for months. The harmful effects of insecticide residues on dung fauna can compromise crucial ecosystem services in terms of grazing, especially in contexts influenced by climate change [19].

Root nodules are structures formed by the roots of leguminous plants in symbiosis with Rhizobium bacteria, which fix atmospheric nitrogen by converting it into forms usable by the plant, thereby improving soil fertility and reducing the need for synthetic fertilizers. However, the efficiency of this symbiosis can be compromised by xenobiotics such as pesticides and environmental pollutants, which interfere with the chemical signals necessary for the initiation and maintenance of symbiosis. These chemicals can delay the recruitment of Rhizobium bacteria, reduce nitrogen fixation, and consequently decrease crop yields, threatening agricultural sustainability [7].

Previous research has demonstrated that deltamethrin significantly affects the growth, pigment levels, carbohydrate content, protein composition, and nitrogen content, in the rice field, of the cyanobacterium *Calothrix* sp. [13]. And other studies have demonstrated, similarly, a gradual but notable decline in the growth rate of nitrogen-fixing cyanobacteria as deltamethrin concentrations increased [20]. However, a direct measurement of the absolute and relative mass of the entire system of nitrogen-fixing nodules is still missing.

The pyrethroid-dependent inhibition mechanism of nitrogen fixation is still unclear [21], but there is considerable evidence that it might interfere directly with nitrogenase’s activity [7].

Our study demonstrated that, at the time of sample collection, the plants growing in the treated field showed significantly fewer absolute nitrogen-fixing nodules (Figure 2), with a reduction of 0.416 FC (Table 5). Conversely, the root system (without nodules) of the plants treated with deltamethrin was consistently bigger, with a FC of 1.32 (Figure 3, Table 6). The entire root system, including the nitrogen-fixing nodules, was still bigger in the deltamethrin-treated group (FC 1.21), even taking into consideration the smaller weight of the nitrogen-fixing nodules.

Legume roots could compensate for poor nitrogen fixation by increasing their growth when the number of root nodules is low. Symbiotic root nodules are essential for providing active cytokinins, which promote shoot growth, underscoring their role in overall plant development [22], and nodule formation is intricately controlled by signal transduction pathways responsive to nitrogen availability, highlighting the critical need for effective nodulation to acquire nitrogen efficiently [23]. Additionally, studies have shown that treating legume seeds with specific biopreparations can significantly boost nodule numbers, leading to enhanced nitrogen accumulation in the soil and increased protein content in the plant herbage, demonstrating the potential for improving nitrogen fixation through external interventions. Therefore, while compensatory root growth can occur, avoiding factors negatively impacting this symbiosis might be essential for efficient nitrogen acquisition in legumes [24].

The obtained results are even more explicative if we take into consideration, rather than the absolute weight, the nodules/roots ratio. As visible in Figure 5, the control group shows a higher ratio of nitrogen-fixing nodules (FC 0.362, *p*-value 0.0001), demonstrating the root system grows independently from nitrogen-fixing nodules, possibly to compensate for the deltamethrin-induced nitrogen-fixing-bacteria’s growth inhibition.

Xenobiotics in general and, more precisely, pesticides were already demonstrated to be detrimental for nitrogen-fixing rhizobia [7] and, in this specific case, deltamethrin demonstrates its role as a negative influence on the abundance of nitrogen-fixing nodules. Moreover, this study suggests a compensatory response, where plants increase their root mass to mitigate the chemical stress induced by the pyrethroid deltamethrin. Plants’ root growth may aim to enhance nutrient and water uptake, potentially counteracting the pesticide’s adverse effects.

The vast majority of biologically fixed N is attributable to symbioses between leguminous plants (soybean, alfalfa, etc.) and species of *Rhizobium* bacteria, and effective nitrogen fixation can significantly reduce the need for synthetic N fertilizers. This is detrimental to crops and soil health and reduces their efficiency in terms of water consumption [7].

In summary, this study suggests that deltamethrin induces a dual response in *Medicago truncatula*: increased root growth to compensate for reduced nodular efficiency, highlighting the plants’ adaptive strategies under chemical stress, while concurrently impairing essential nitrogen-fixing processes crucial for plant productivity and health.

The negative effect of deltamethrin on nitrogen-fixing nodules is consistent with previous studies showing that pesticides can damage soil microbial communities and their activity [7,8,9,10,11,12,13,14,15,16,17,18,19,20,21,22,23,24,25]. This suggests that deltamethrin not only acts as an insecticide but also compromises microbiological processes essential for ecosystem health.

The negative ecological impact highlighted by this study underscores the urgency of adopting more sustainable and environmentally friendly agricultural management practices. Reducing the use of chemical pesticides and adopting sustainable alternatives [26,27], such as biological control and natural treatments, can help protect ecosystem services such as biological nitrogen fixation, contributing to biodiversity conservation and long-term agricultural sustainability.

Finally, it is important to re-consider the use of deltamethrin in livestock, where it is employed as an antiparasitic treatment for grazing animals. To prevent this insecticide from contaminating the soil and compromising ecosystem health, it would be advisable not to treat grazing animals with these molecules [6]. The adoption of “Green Veterinary Pharmacology” can offer safer and more sustainable alternatives, as this discipline promotes the use of natural and biological treatments that reduce environmental impact and contribute to the conservation of pastoral ecosystems while ensuring the health and well-being of animals [28].

## 5. Conclusions

This study demonstrates that deltamethrin, a commonly used pyrethroid in livestock farming, significantly reduces both the number and biomass of nitrogen-fixing nodules in *M. truncatula*. This negative effect on the symbiotic relationship between the plant and nitrogen-fixing bacteria could impair soil fertility and plant growth. These findings highlight the ecological risks associated with deltamethrin’s use and emphasize the importance of adopting Green Veterinary Pharmacology practices. By reducing our reliance on harmful chemicals, environmentally sustainable practices can protect essential ecosystem services such as biological nitrogen fixation, which is crucial for maintaining soil health and agricultural productivity.

To mitigate the risks associated with deltamethrin, it is beneficial to consider the use of plant extracts, which can provide ecological and sustainable alternatives for pest control. Future research should explore the effects of other common pesticides on nitrogen-fixing nodules and develop strategies to mitigate their impacts. Additionally, investigating the potential of plant extracts to enhance agricultural sustainability and soil health could provide valuable insights for optimizing pest management practices in an environmentally friendly manner.

## Figures and Tables

**Figure 1 toxics-12-00615-f001:**
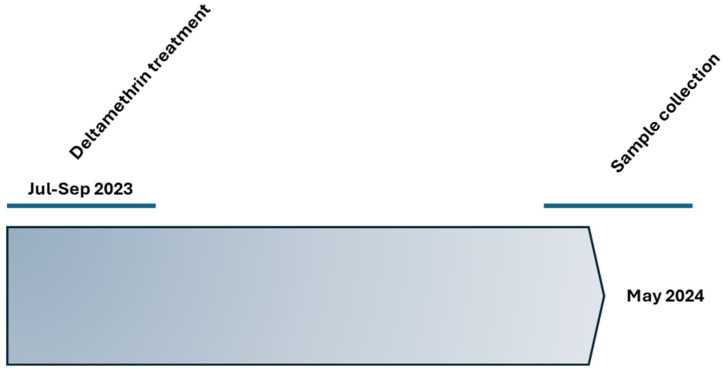
Timeline of the experiments and sample collection.

**Figure 2 toxics-12-00615-f002:**
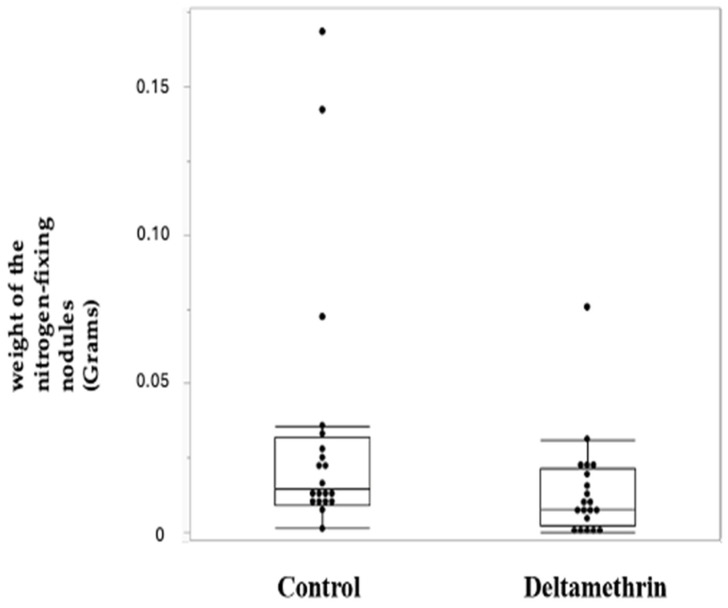
Box plot reporting the weight of the nitrogen-fixing nodules in the control and the deltamethrin-treated group.

**Figure 3 toxics-12-00615-f003:**
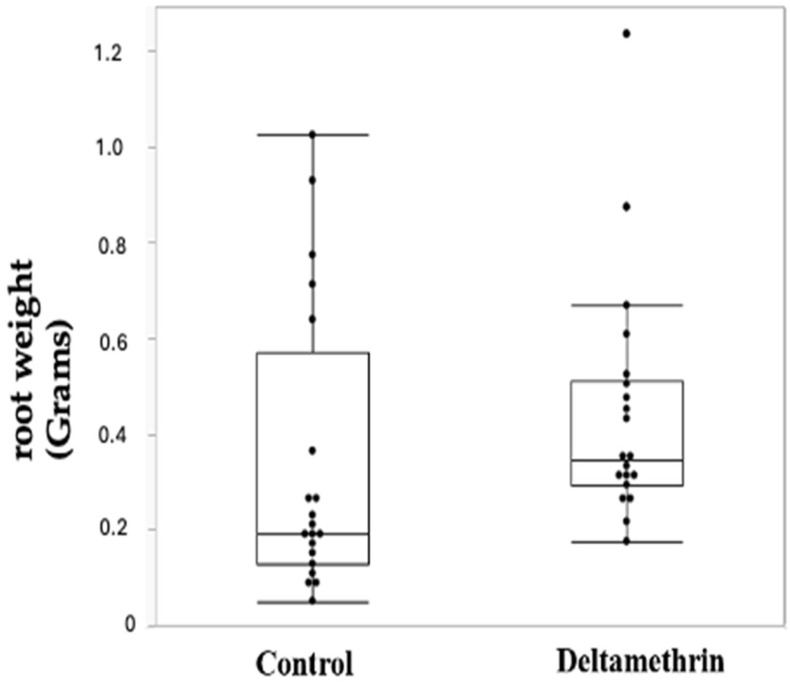
Box plot reporting the root weights among the control and the deltamethrin-treated group.

**Figure 4 toxics-12-00615-f004:**
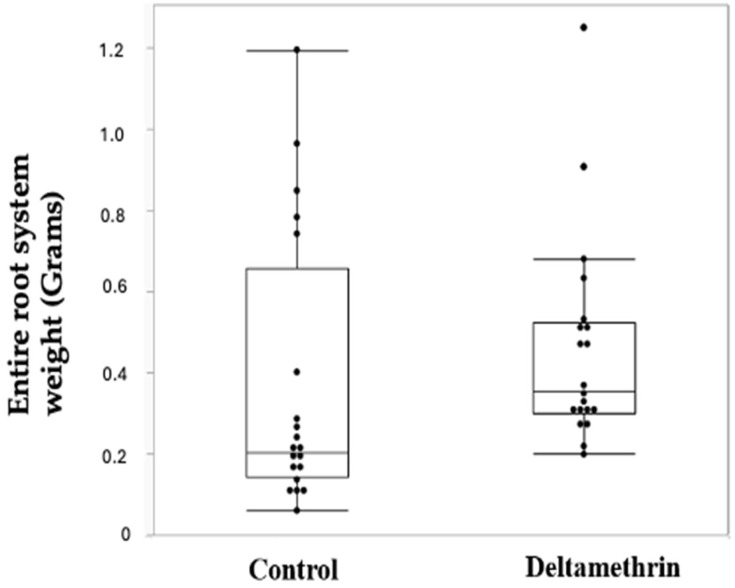
Box plot of the entire root system weights of both the control and the deltamethrin-treated group.

**Figure 5 toxics-12-00615-f005:**
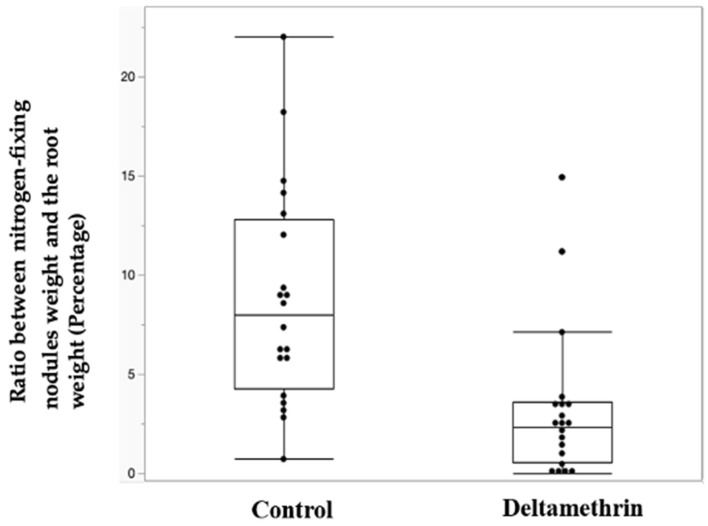
Box plot reporting the ratio between nitrogen-fixing nodules’ weight and the root weight.

**Table 1 toxics-12-00615-t001:** Daily precipitation measurements in Lamezia Terme from July 2023 onwards. This table details the amount of precipitation recorded each month after the commencement of the deltamethrin treatment.

July	August	September	October	November	December
24 mm	30 mm	84 mm	104 mm	110 mm	91 mm

**Table 2 toxics-12-00615-t002:** Maximum temperature values in Lamezia Terme from July 2023 onwards. This table presents the daily maximum temperatures recorded since the start of the deltamethrin treatment.

July	August	September	October	November	December
27 °C	27 °C	24 °C	21 °C	17 °C	13 °C

**Table 3 toxics-12-00615-t003:** Minimum temperature values in Lamezia Terme from July 2023 onwards. This table displays the daily minimum temperatures recorded following the initiation of the deltamethrin treatment.

July	August	September	October	November	December
21 °C	22 °C	19 °C	16 °C	12 °C	9 °C

**Table 4 toxics-12-00615-t004:** Soil characteristics of the study area. This table outlines the pH levels and the percentages of silt, sand, and clay in the soil.

pH	Soil Type	Silt	Sand	Clay
6.8	Clayey	35.55	17.37	47.08

**Table 5 toxics-12-00615-t005:** Mean values of the nitrogen-fixing nodules in the control and the deltamethrin-treated group (* Kruskal–Wallis test).

Mean Control	Mean Tread	Fold Change	Prob > Z *
0.033	0.014	0.416	0.0103

**Table 6 toxics-12-00615-t006:** Mean values of root weight in the control and the deltamethrin-treated group. (* Kruskal–Wallis test).

Mean Control	Mean Treted	Prob > Z *	Fold Change
0.335	0.444	0.0104	1.324

**Table 7 toxics-12-00615-t007:** Mean values of the entire root system weight of both the control and the deltamethrin-treated group. (* Kruskal–Wallis test).

Mean Control	Mean Treated	Fold Change	Prob > Z *
0.377	0.458	1.216	0.0096

**Table 8 toxics-12-00615-t008:** Mean values of the ratio between the nitrogen-fixing nodules’ weight and the root weight. (* Kruskal–Wallis test).

Mean Control	Mean Tread	Fold Change	Prob > Z *
8.739	3.167	0.362	0.0001

## Data Availability

The data are contained within the Appendix A.

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
