# Peer review of "Deltamethrin’s Effect on Nitrogen-Fixing Nodules in Medicago truncatula"

_toxics, 2024, doi:10.3390/toxics12080615_

Round 1
Reviewer 1 Report
Comments and Suggestions for Authors
The manuscript investigates the impact of deltamethrin, a pyrethroid insecticide, on the nitrogen-fixing nodules of M. truncatula. The study concludes that deltamethrin significantly reduces the biomass of these nodules, which could impair soil fertility and plant growth. The results suggest a need for environmentally sustainable practices in managing livestock health.
The study addresses an important ecological concern regarding the widespread use of deltamethrin in livestock farming. The focus on nitrogen-fixing nodules in M. truncatula is novel and relevant for understanding the broader environmental impacts of pyrethroids.
This MS presents essential findings relevant to the field of environmental toxicology. With some revisions to enhance the mechanistic insights, methodological details, and discussion, it would be suitable for publication in Toxics.
- The paper provides a clear link between deltamethrin exposure and reduced nodule biomass; however, further mechanistic details on how deltamethrin affects these bacteria would strengthen the manuscript.
- The introduction provides good overview of this issue but could be improved by briefly discussing previous studies on the effects of pesticides on nitrogen-fixing bacteria.
- Clearly state the hypothesis or research question at last paragraph.
- it would be beneficial to mention the soil type and other relevant environmental conditions (e.g., temperature, rainfall) that might affect the results.
- Ensure that the storage conditions (-20°C) are appropriate for maintaining sample integrity.
- Figure 1: There is nothing special in Fig. 1 and I think it should be deleted
- The discussion could be strengthened by comparing the findings with previous studies on similar topics.
- How does deltamethrin specifically interfere with the nitrogen-fixing process? Are there any known pathways or cellular targets?
- Consider the broader implications of these findings. How might reduced nodule biomass affect overall plant health and soil ecosystems in the long term?
- The conclusion summarizes the findings well but could be more concise. Reiterate the importance of adopting Green Veterinary Pharmacology practices and suggest specific alternatives to deltamethrin.
- Provide suggestions for future research, such as exploring the impact of other common pesticides on nitrogen-fixing nodules or investigating mitigation strategies.
- Check for any grammatical errors and improve sentence structure for clarity.
- For mor comments, plz check uploaded file.

Minor editing of English language required
Author Response
The introduction provides good overview of this issue but could be improved by briefly discussing previous studies on the effects of pesticides on nitrogen-fixing bacteria.
Response: Thank you for the suggestion that helped to improve the introduction section making more clear the aim of this study. We have incorporated a brief discussion of previous studies on the effects of pesticides on nitrogen-fixing bacteria into the introduction. We believe this addition enriches the context and provides a more comprehensive view of the issue at hand.
Clearly state the hypothesis or research question at last paragraph.
Response: Thank you for the comment! We have made the changes, and now the hypothesis or research question is clearly stated in the last paragraph.
it would be beneficial to mention the soil type and other relevant environmental conditions (e.g., temperature, rainfall) that might affect the results.
Response: Thank you for your suggestion. We apologize for not initially including the details regarding soil type and environmental conditions such as temperature and rainfall, which are certainly relevant to the results. Preliminary soil analyses had been conducted, but they were not specified in the data presented. We have now added the requested information to the "Results" section of our manuscript.
Ensure that the storage conditions (-20°C) are appropriate for maintaining sample integrity.
Response: The samples were only temporarily stored at 20 (max 24h) and, subsequently transferred at -80. For the analysis we conducted, which was mass control, the temporary storage conditions at -20°C were suitable to keep the results reliable. This part has now been better described in the methods section.
Figure 1: There is nothing special in Fig. 1 and I think it should be deleted
Response: Thank you for your comment. As this figure was commented by the reviewer number 2 as well, in agreement with his comment, we moved the image to the Methods section.
The discussion could be strengthened by comparing the findings with previous studies on similar topics.
Response: Many thanks for this really pertinent comment that helped us to give a clearer view to the reader. The discussion section has now been implemented with a new part (lines 254-262).
How does deltamethrin specifically interfere with the nitrogen-fixing process? Are there any known pathways or cellular targets?
Response: The mechanism of explication of this effect still needs to be fully understood, however, it seems that the nitrogenase inhibition might be one of the key factors. This point has now been implemented in lines 263-266 of the discussion section.
Consider the broader implications of these findings. How might reduced nodule biomass affect overall plant health and soil ecosystems in the long term?
Response: This represents a very important point to be underlined in the presentation of our results. A paragraph was added in the discussion section: “The vast majority of biologically fixed N is attributable to symbioses between leguminous plants (soybean, alfalfa, etc.) and species of Rhizobium bacteria and, effective nitrogen fixation can significantly reduce the need for synthetic N fertilizers. This is detrimental to crops and soil health and reduces efficiency in water consumption” On behalf of all authors, I would like to thank the referee for this comment that was necessary to better outline the overall picture.
The conclusion summarizes the findings well but could be more concise. Reiterate the importance of adopting Green Veterinary Pharmacology practices and suggest specific alternatives to deltamethrin. Provide suggestions for future research, such as exploring the impact of other common pesticides on nitrogen-fixing nodules or investigating mitigation strategies.
Response: Thank you for your feedback. We have revised the conclusion to make it more concise and focused.
Check for any grammatical errors and improve sentence structure for clarity.
Response: Thank you for the suggestion. We have made the changes you indicated.
Reviewer 2 Report
Comments and Suggestions for Authors
Dear Editor and Authors
The manuscript presents an evaluation of the deltamethrin effect on nitrogen-fixing nodules in Medicago truncatula. Although it is a very simple evaluation, lacking information, for example, from molecular analyses, I consider the results to be interesting. However, the context is not clear. The manuscript presented that the problem is related to the use of pyrethroid in animal production, but the variable tested was the application of the insecticide in an olive grove. In fact, this relationship is confusing.
The text from the first paragraph to Figure 1 of the results should be transferred to the Material and methods section;
The number of samples in both areas was not presented;
The legend for the Y axes of all figures is missing.
Author Response
The manuscript presents an evaluation of the deltamethrin effect on nitrogen-fixing nodules in Medicago truncatula. Although it is a very simple evaluation, lacking information, for example, from molecular analyses, I consider the results to be interesting. However, the context is not clear. The manuscript presented that the problem is related to the use of pyrethroid in animal production, but the variable tested was the application of the insecticide in an olive grove. In fact, this relationship is confusing.
Response: Thank you for your feedback on our manuscript evaluating the effect of deltamethrin on nitrogen-fixing nodules in Medicago truncatula. We appreciate your interest in our results.
To clarify the context, the choice of an olive grove as our study model was intentional. The use of deltamethrin in such settings is relevant due to its application for controlling various pests that affect olive trees, including Saissetia oleae, Prays oleae, Bactrocera oleae, Aromia bungii, and Philaenus spumarius. Our intent was to simulate potential environmental impacts, specifically the scenario where deltamethrin could enter the soil through animal manure, as this pesticide is also used in animal production for pest control. Thus, by using the olive grove as a model, we aimed to assess how deltamethrin residues might affect soil health and nodulation in a context that reflects both agricultural and animal production practices.
Moreover this was better specified at the beginning of the discussion section in lines 227-229.
We hope this explanation clarifies the rationale behind our experimental design. Thank you again for your valuable input.
The text from the first paragraph to Figure 1 of the results should be transferred to the Material and methods section;
Response: Thank you for your comment. We have moved Figure 1 from the Results section to the Materials and Methods section, as requested.
The number of samples in both areas was not presented;
Response: We apologize for not including the number of samples in the specified areas. We have updated the Materials and Methods section to include this information. Thank you for your understanding.
The legend for the Y axes of all figures is missing.
Response: Thank you for the comment! The legends for the Y axes have been added to all the figures.
Reviewer 3 Report
Comments and Suggestions for Authors
Dear Editor,
Enclosed please find the comments on the following manuscript:
Manuscript ID: toxics-3128526
Type of manuscript: Communication
Title: Deltamethrin effect on Nitrogen-Fixing Nodules in Medicago Truncatula
The manuscript “Deltamethrin effect on Nitrogen-Fixing Nodules in Medicago Truncatula” by Rosario De Fazio et. al, is a communication article in a current and interesting topic and could be of interest for veterinary and agriculture scientists. Authors used deltamethrin to significantly reduce the biomass of nitrogen-fixing bacteria in the roots of Medicago truncatula crops. The manuscript is well written, the methodology and results clearly described and discussed. The results reported are of interest for the scientific community. This research may contribute to the veterinary pharmacology and agriculture.
1. First, authors missed the completely randomized design (CRD) or completely randomized block design (CRBD) and samples numbers in Materials and Methods.
2. Authors should address the negative mechanisms between control and Deltamethrin group for this field trial.
3. Authors should correct crop scientific name format.
4. Page 3, line 101: M. truncatula.
5. Page 8, line 229: Medicago truncatula.
Author Response
First, authors missed the completely randomized design (CRD) or completely randomized block design (CRBD) and samples numbers in Materials and Methods.
Response: Thank you very much for your comment. We have made a revision to the manuscript to clarify the point you raised (lines 98-104). we hope the update makes everything clearer.
Authors should address the negative mechanisms between control and Deltamethrin group for this field trial.
Response: Thank you sincerely for your constructive and valuable comments regarding our manuscript. Your observations have been very helpful in improving the quality of our work. We would like to inform you that we have made the suggested revisions in lines 54 to 68.
Authors should correct crop scientific name format.
Response: Thank you for your comment. All scientific names have been checked and are correct.
Page 3, line 101: M. truncatula. Page 8, line 229: Medicago truncatula.
Response: Thank you for the suggestion, the changes have been made.
Round 2
Reviewer 1 Report
Comments and Suggestions for Authors
now it can be accepted.
Comments on the Quality of English LanguageMinor editing of English language required.
Author Response
comment:now it can be accepted.
Response:Thank you very much for your comment and for the time you dedicated.
Reviewer 2 Report
Comments and Suggestions for Authors
Dear Editor and Authors
The manuscript continues to place too much emphasis on manure as a source of pyrethroids for soils. For example, the abstract and introduction address this issue. However, the manuscript does not include areas where manure is used, but rather olive groves. It makes no sense to place so much emphasis on the issue of manure; I suggest that this topic be mentioned in a secondary way, since the study evaluates the impact on olive groves.
Author Response
Comment: The manuscript continues to place too much emphasis on manure as a source of pyrethroids for soils. For example, the abstract and introduction address this issue. However, the manuscript does not include areas where manure is used, but rather olive groves. It makes no sense to place so much emphasis on the issue of manure; I suggest that this topic be mentioned in a secondary way, since the study evaluates the impact on olive groves.
Response: On behalf of all authors, I would like to thank the referee for the previous revision and for this last comment. It is indeed correct to underline the non-animal purposes of deltamethrin uses. The abstract and the introduction have now been amended.
Reviewer 3 Report
Comments and Suggestions for Authors
This revised manuscript can be accepted for publication.
Author Response
comment:This revised manuscript can be accepted for publication.
response:Thank you very much for your comment and for the time you dedicated.
Round 3
Reviewer 2 Report
Comments and Suggestions for Authors
The Y-axis measurement units of Figures 2, 3, 4 and 5 must be included.
Author Response
Comment:The Y-axis measurement units of Figures 2, 3, 4 and 5 must be included.
Response: Thank you for your comment. We have added the units of measurement to figures 2, 3, and 4. As for figure 5, it represents a percentage.